# Quantitative Risk Assessment of Hepatitis a Virus Infection Arising from the Consumption of Fermented Clams in South Korea

**DOI:** 10.3390/foods12040796

**Published:** 2023-02-13

**Authors:** Yoonjeong Yoo, Miseon Sung, Jeongeun Hwang, Daseul Yeo, Ziwei Zhao, Changsun Choi, Yohan Yoon

**Affiliations:** 1Department of Food and Nutrition, Sookmyung Women’s University, Seoul 04310, Republic of Korea; 2Department of Food and Nutrition, College of Biotechnology and Natural Resources, Chung-Ang University, Seoul 17546, Republic of Korea

**Keywords:** fermented clam, hepatitis A virus, quantitative microbial risk assessment (QMRA), foodborne illness, predictive model

## Abstract

This study estimated the risk of hepatitis A virus (HAV) foodborne illness outbreaks through the consumption of fermented clams in South Korea. HAV prevalence in fermented clams was obtained from the Ministry of Food and Drug Safety Report, 2019. Fermented clam samples (2 g) were inoculated with HAV and stored at −20–25 °C. Based on the HAV titer (determined using plaque assay) in fermented clams according to storage, the Baranyi predictive models provided by Combase were applied to describe the kinetic behavior of HAV in fermented clams. The initial estimated HAV contamination level was −3.7 Log PFU/g. The developed predictive models revealed that, when the temperature increased, the number of HAV plaques decreased. The Beta-Poisson model was chosen for determining the dose–response of HAV, and the simulation revealed that there was a 6.56 × 10^−11^/person/day chance of contracting HAV foodborne illness by eating fermented clams. However, when only regular consumers of fermented clams were assumed as the population, the probability of HAV foodborne illness increased to 8.11 × 10^−8^/person/day. These results suggest that, while there is a low likelihood of HAV foodborne illness from consuming fermented clams across the country, regular consumers should be aware of the possibility of foodborne illness.

## 1. Introduction

The hepatitis A virus (HAV) is an RNA virus belonging to the genus *Hepatovirus* within the family *Picornaviridae.* HAV infects the blood through the intestinal tract and causes liver inflammation. There are seven HAV genotypes (I [A, B], II, III [A, B], IV, V, VI, and VII), of which IA and IB genotypes mainly infect humans [1]. Upon infection with HAV, clinical symptoms appear after an incubation period of 4 weeks. HAV infection can cause fever, loss of appetite, vomiting, abdominal pain, diarrhea, and jaundice in adults. HAV can be transferred by the fecal–oral route; however, the majority of cases involve direct human-to-human contact or indirect transmission through the consumption of contaminated food or drinks. It is estimated that 10% of hepatitis A infections in the US are caused by food, and these outbreaks have a significant financial impact on both society and the food industry [2,3,4].

HAV is known as a causative agent of foodborne illnesses in several countries, and mainly contaminates clams such as oysters and mussels, and fruits and vegetables such as leafy vegetables and berries [5]. Since bivalves accumulate various viruses in their midgut through filtration, shellfish, such as oysters and clams, are the main causes of viral foodborne illness [6,7]. Accordingly, research on contaminated food, as well as case studies of patients infected with HAV are required. According to the World Health Organization, more than 100 million HAV infections occur annually [8]. In 2019, hepatitis A infection was prevalent in Korea, and the number of patients was 17,598, a sevenfold increase compared to the previous year (2,437 patients in 2018) [9]. The main cause was identified as HAV-contaminated fermented clams called Jogaejeotgal in South Korea [10]. Fermented clams are a fermented food that is self-digested and matured by adding salt to fresh clams to prevent spoilage [11]. The salinity of the fermented clams is 9.6%, and the pH ranges from 5.5 to 6.0 [12,13].

Quantitative microbial risk assessment (QMRA), a scientific method for quantitatively estimating the risk caused by microorganisms, entails hazard identification, exposure assessment, hazard characterization, and risk characterization [14].

In this study, the risk of HAV by intake of fermented clams in South Korea was estimated with QMRA and the results are expected to aid in risk management.

## 2. Materials and Methods

### 2.1. Estimation of Initial Contamination Concentration

In 2019, the Ministry of Food and Drug Safety (MFDS) analyzed fermented clam samples to detect HAV in South Korea, and the data were cited in this study. The prevalence data were fitted with the Beta distribution, and @RISK software ver.8.0 (Palisade Corp., Ithaca, NY, USA) was used to estimate the initial contamination level [15,16].

### 2.2. Estimation of Time and Temperature Data during Transport and Storage of the Fermented Clams

Through personal communication with market employees and literature reviews, the temperature and time data during transport and storage were collected. The data were then used to simulate the fate of HAV using the models developed in this study.

### 2.3. HAV Inoculation and Titer Enumeration in the Fermented Clams

After grinding a fermented clam sample (25 g) for 1 min, 2 g of each sample was transferred to a 50 mL tube. Fermented clam samples were inoculated with 300 µL of HAV HM175 (VR-2097; ATCC, Manassas, VA, USA) to obtain 7.4 Log PFU/mL. The HAV-inoculated fermented clam samples were then incubated for 0, 1, 2, 3, 5 and 7 days at −20 °C, 4 °C, and 25 °C. The artificially inoculated HAV in the samples was recovered using the next steps. The fermented clam samples were treated with 18 mL glycine buffer (0.1 M glycine, 0.3 M NaCl; pH 9.5; Sigma-Aldrich, St. Louis, MO, USA). To obtain the supernatant, the suspension was vortexed at 21 °C for 5 min before centrifugation(Beckman Coulter, Brea, CA, USA) at 10,000× *g* at 4 °C for 15 min. The supernatant was treated with 20 mL of 16% polyethylene glycol (PEG; Sigma-Aldrich) 8000 containing 0.25 M NaCl and then rocked for 20 min (60 oscillations/min) at room temperature. The supernatant was discarded after the suspension was centrifuged for 10 min at 10,000× *g* at 4 °C. Then, 1 mL of phosphate-buffered saline was used to resuspend the pellet (pH 7.0). The resultant solution was diluted 10 times in Hank’s balanced salt solution (Gibco, Carlsbad, CA, USA) and filtered through 0.22 μm and 0.45 μm syringe filters. The filtrate was immediately assayed for plaque formation. Plaque assays for HAV were performed as previously described [17]. Fetal Rhesus Kidney-4 (FRhK-4) cells were seeded approximately 2 × 10^4^ cells/well in 12-well plates and incubated at 37 °C and 5% CO_2_ for 4–6 days to reach 95% confluence for each well. The inoculum extracted from the fermented clam samples was serially diluted 10-fold in serum-free Dulbecco’s Modified Eagle Medium (DMEM, Gibco). Each well of confluent FRhK-4 cells was inoculated with 0.1 mL of diluted inoculum, and the plates were incubated for 1 h at 37 °C and 5% CO_2_. After incubation, 1 mL of type 2 agarose mixed with 2 × DMEM (1:1) was overlaid at room temperature and incubated for 6–7 days at 37 °C and 5% CO_2_. Cells were fixed with 10% neutral formalin (Sigma-Aldrich) and stained for 20 min at room temperature with a 0.1% crystal violet solution. The HAV titer was calculated as Log PFU/g [18]. 

### 2.4. Predictive Model Development

To develop primary models, the HAV plaque counts were fitted to the Baranyi model with DMfit 3.5, provided by Combase, as follows [15]:$$N_{t}=N_{0}+A_{t}\times DR-ln\left[1+\frac{\exp\left(DR\times A_{t}\right)-1}{\exp\left(N_{max}-N_{0}\right)}\right]$$
$$A_{t}=t+\frac{1}{DR}ln\left(\frac{\exp\left(-DR\right)+q_{0}}{1+q_{0}}\right)$$
$$q_{0}=\frac{1}{\exp\left(h_{0}\right)-1}$$
where *N_max_* and *N_0_* denote the final and initial viral titers, respectively, *N_t_* denotes the viral titers at time *t*, and *h_0_* is a parameter that defines the initial physiological state of HAV. *DR* (Log PFU/g/h) represents the death rate of HAV, *A_t_* represents the physiological state of the virus titer when the shoulder period is defined as the time in hours where the levels of pathogen remain at the level of inoculation and *q_0_* is a measure of the initial physiological state of the cells [19]. The Baranyi model estimates HAV quantity after exposure to various temperatures along the three different counting points, market, transportation, and home storage. Because HAV does not grow in food, the Baranyi model in the death curve model was used based on two parameters: shoulder period (h) and death rate (Log CFU/g/h). To assess the effect of temperature on *DR*, *DR* data were fitted with the second predictive model described below. *DR* = *Y*_0_ + *a* × *T*
where *Y*_0_ is constant, *a* is the rate constant, and *T* denotes temperature. 

Fermented clams were infected with HAV as described above and stored at 15 °C to validate the efficacy of the predictive models with the parameter estimates in the Baranyi model. To obtain the observed values, HAV plaque counts were determined during storage using the plaque assay described above. The root mean square error (*RMSE*), accuracy factor (*A_f_*), and bias factor (*B_f_*) were calculated to measure the differences between the observed and predicted HAV titers, as follows [20]: $$\mathrm{RMSE}=\sqrt{\sum {\left(observed\;value-predicted\;value\right)}^{2}/n}$$
$$A_{f}=10^{\left(\sum \left|Log\left(predicted\;value/observed\;value\right)\right|/n\right)}$$
$$B_{f}=10^{\left(\sum Log\left(predicted\;value/observed\;value\right)/n\right)}$$
where *n* means the number of data points.

### 2.5. Consumption Ratio and Amount of Fermented Clams in South Korea 

The consumption amounts and ratio of fermented clams in South Korea were calculated using data from the Korea Disease Control and Prevention Agency (KDCA) Korea National Health and Nutrition Examination Survey (KNHANES) in 2018. The consumption amount and ratio of the fermented clams were extracted from the raw data using SAS^®^ version 9.4 (SAS Institute Inc., Cary, NC, USA), and duplicate data were excluded by performing a redundancy test in an Excel spreadsheet. The consumption data were evaluated, and an acceptable probability distribution was determined using @RISK software.

### 2.6. Dose–Response Model

The HAV dose–response model was investigated using published studies to analyze the response to HAV foodborne illness after the consumption of fermented clams.

### 2.7. Risk Characterization

In an Excel spreadsheet, a simulation model was developed using the HAV level of contamination in fermented clams, predictive models, probabilistic distributions for temperature and time during transportation and storage, probabilistic distribution of consumption amount, consumption ratio, and dose–response model. This simulation model was used for the simulation using @RISK with 10,000 iterations to calculate the risk of HAV foodborne illnesses caused by fermented clam ingestion per person per day. Furthermore, variables with a major impact on risk were identified.

## 3. Results and Discussion

### 3.1. Prevalence and Initial Contamination Level of HAV in Fermented Clams

The investigation of HAV contamination in fermented clams revealed that 44 of 136 samples were positive for HAV [21]. Of these 44 samples, 30 were produced in South Korea and 14 were imported. This result was fitted to the Beta distribution [Beta(45, 93)], and the initial contamination level of HAV in the fermented clams was estimated using @RISK. The average level was −3.7 Log PFU/g. The prevalence of pathogenic microorganisms in salt-treated foods is thought to be rare; however, as evidenced by recent HAV foodborne illness cases of fermented clams in South Korea, numerous microorganisms have been discovered in foods with high concentrations of salt [21,22,23]. While research on microbial contamination of meat such as chicken, cattle, and pork has been conducted sufficiently, research on microbial contamination of aquatic products containing salt, sugar, and acid is limited [24,25,26,27,28]. Therefore, investigations on the microbiological contamination level of marine products that contain salt, sugar, and acid should be carried out continually; in particular, studies on the contamination of viruses that cause foodborne illnesses need to be conducted.

### 3.2. Temperature and Time of the Fermented Clams during Storage and Transportation

To predict the number of HAV immediately before consumption, the probabilistic distributions of the fermented clams were set as moving from the market to home storage and consumption. The temperature displayed in the markets was fitted to the Uniform distribution [Uniform(2.2281, 35)] [29,30]. The display time at the market showcase was up to 5 months and was mainly displayed for 1 week. Hence, it was fitted to the Pert distribution [Pert(0, 168, 3600)]. For the time and temperature at which consumers purchase food from a market and move to their homes, data from Jung (2011) were used [30]. The transportation time ranged from 0.325 h to 1.643 h, and thus, it was fitted to the Uniform distribution [Uniform(0.325, 1.643)]. The temperatures during the fermented clams transportation ranged from 10 °C to 25 °C; thus, it was fitted to the Pert distribution [Pert(10.0, 18.0, 25.0)]. For home storage, fermented clams were stored for up to 720 h [31]. Thus, it was fitted to the Uniform distribution [Uniform(0, 720)]. The LogLogistic distribution [LogLogistic(−29.28, 33.22, 26.66, Truncate(−5, 10)] from a study by Lee et al. (2015) was referenced for the storage temperature at home [32].

### 3.3. Predictive Models to Describe the Fate of HAV during Transport and Storage as Temperature and Time Changes

To describe the changes in the HAV contamination level in the fermented clams according to the changes in temperature and time, predictive models were developed. The generated curves of primary predictive models for the number of HAV with the Baranyi model based on temperature and time changes revealed that viral titers at 25 °C decreased slightly over time. At −20 °C and 4 °C, the HAV concentration barely changed (Figure 1). 

Accordingly, the shoulder period of HAV for fermented clams was 70 h at −20 °C, but the shoulder period was 5.4 h at 25 °C (Table 1). The salinity of fermented clams is approximately 10%, which may not interact with the virus at low temperatures. However, at 25 °C, 10% salinity may interfere with the molecular structure of the virus [33]. Thus, a slight decrease in the HAV titer was observed at 25 °C. As the temperature increased, the shoulder period decreased (Table 1). Secondary predictive models were created to evaluate the influence of temperature on the shoulder period and death rate values using a linear model (Figure 2). The developed secondary models and their corresponding graphs are shown in Figure 2. An additional experiment was conducted at 15 °C to validate the performance of the developed predictive models. Although the *R^2^* value of the developed secondary model was not high, the validation results showed that the developed predictive models were applicable to depict the fates of HAV in fermented clams. The *RMSE* value was 0.116, which was close to 0, and *A_f_* and *B_f_* were 1.00 and 1.03, respectively, which were close to 1. 

The changes in the HAV contamination level in the fermented clams simulated by the developed predictive models with the probabilistic distributions of temperature and time during transport and storage are presented in Figure 3. When the initial contamination level went through the market (C1)—transfer from market to house (C2)—storage at the house (C3), the average predicted HAV contamination level slightly decreased from −6.7 to −7.8 Log PFU/g.

### 3.4. Consumption Ratio and Amount of Fermented Clams

Nine participants stated they ate ‘fermented clam (Jogaejeotgal)’ out of a total of 7064 who took the KNHANES in 2018. Accordingly, 0.13% of South Koreans consumed fermented clams, and they consumed 9.77 g on average. The optimal probability distribution for the consumption amount was the Exponential distribution [Expon(8.1789, Shift(0.69235))] (Figure 4).

### 3.5. Dose–Response Model

In this study, the Beta Poisson model [P = 1 − (1 + dose/*β*)^−*α*^] with *α* = 0.373 and *β* = 186.4 was cited because this dose–response model was most frequently used in other studies [34].

### 3.6. Risk Characterization and Simulation

To assess the risk of HAV foodborne illness occurring per person per day when fermented clams were ingested, a simulation model was developed using the data gathered in this investigation (Table 2). The results showed that the mean probability of HAV foodborne illness per person per day was 6.56 × 10^−11^ (minimum 0, maximum 2.54 × 10^−7^). Compared to other foods, fermented clams had a lower consumption ratio (0.13%) and average intake (9.77 g), which had an impact on the risk calculation results. Thus, the risk was estimated to be slightly lower. However, when assuming regular consumers of fermented clams, the probability of HAV foodborne illness increased to 8.11 × 10^−8^ on average, with a maximum of 7.12 × 10^−5^. After analyzing the correlation between the variables and risk, the consumption ratio showed the greatest correlation with risk (Figure 5). As a result, while the risk of HAV foodborne illness from fermented clams is low throughout the country, regular consumers of fermented clams should be aware of foodborne illness.

## 4. Conclusions

The risk of HAV foodborne illness from fermented clams in South Korea was 6.56 × 10^−11^/person/day for the entire population, but only for those who consumed the fermented clams, the probability of HAV foodborne illness increased to 8.11 × 10^−8^/person/day. In conclusion, the risk of HAV foodborne illness caused by fermented clams is considered minor throughout the country, but those who regularly consume fermented clams should be aware of this foodborne illness. The consumption ratio might be a highly related factor, which is also shown by the correlation coefficients. Therefore, risk management should be applied to control consumption ratios.

## Figures and Tables

**Figure 1 foods-12-00796-f001:**
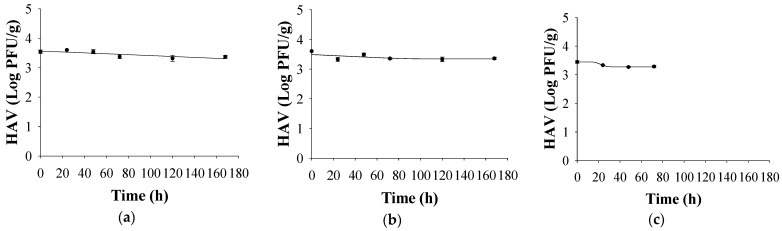
Hepatitis A virus death in the fermented clam during storage at −20 °C (**a**), 4 °C (**b**), and 25 °C (**c**) with the Baranyi model (•: observed data; ―: fitted line).

**Figure 2 foods-12-00796-f002:**
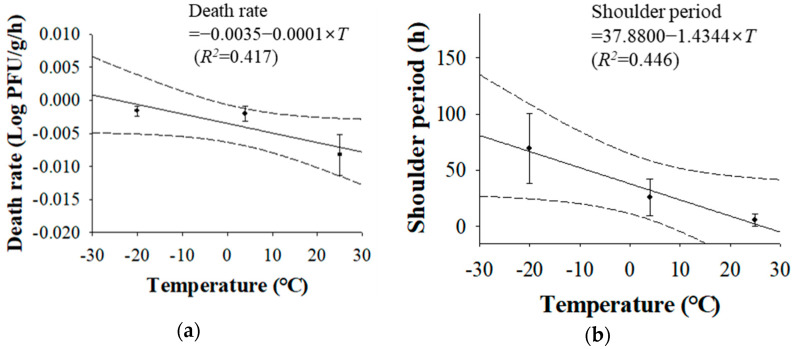
Secondary predictive models for hepatitis A virus death rate (**a**) and shoulder period (**b**) in fermented clams by temperature (*T*: temperature; •: observed data; ―: fitted line).

**Figure 3 foods-12-00796-f003:**
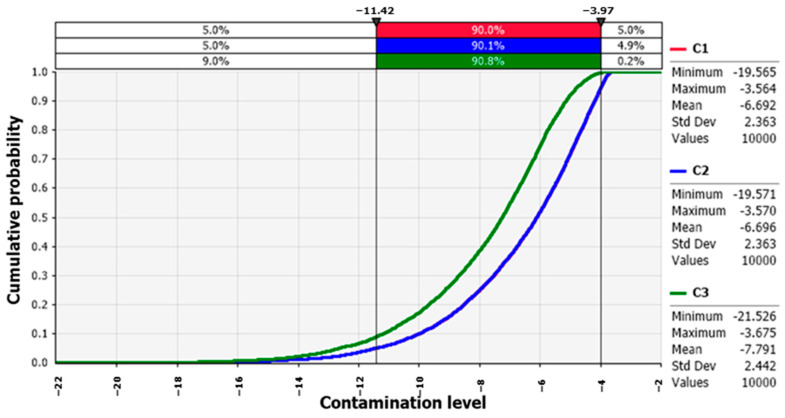
Changes in hepatitis A virus contamination level in the fermented clams according to transportation and storage.

**Figure 4 foods-12-00796-f004:**
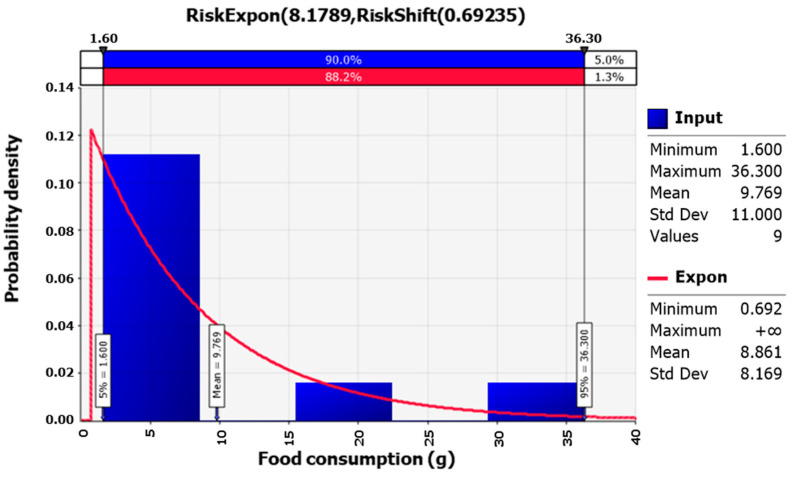
Probability distribution for fermented clam intake from the Korea Health and Nutrition Examination Survey in 2018.

**Figure 5 foods-12-00796-f005:**
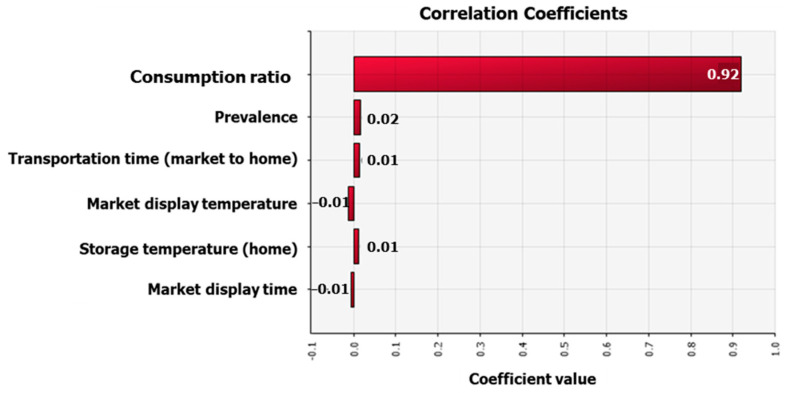
Correlation coefficient values for risk factors influencing the probability of hepatitis A virus-related foodborne illness caused by fermented clam consumption.

**Table 1 foods-12-00796-t001:** Kinetic parameters for hepatitis A virus in fermented clams calculated from the primary predictive model.

Parameter	Temperature		
	−20 °C	4 °C	25 °C
Death rate (Log PFU/g/h)	−1.6 × 10^−3^	−2.0 × 10^−3^	−8.2 × 10^−3^
Shoulder period (h)	70	26	5.4

**Table 2 foods-12-00796-t002:** Simulation model for estimating the risk of hepatitis A virus in the fermented clams with @RISK.

Input	Unit	Variable	Equation	Reference
Initial contamination level				
Hepatitis A virus prevalence		PR	=Beta(45, 93)	This research; [21]
Hepatitis A virus concentration	GC/g	HAV	=−LN(1 − PR)/2 g	[15]
Initial contamination level	PFU/g	CL	=1/1000 × HAV	[35]
	Log PFU/g	LogIC	=Log(CL)	
MARKET				
Display time	h	Time_mark_	=Pert(0, 168, 3600)	This research
Temperature during display	°C	Temp_mark_	=Uniform(2.2281, 35)	[29,30]
Hepatitis A virus death				
		h_0_	=0.001	This research; [19]
	Log PFU/g	Y_0_	=average(Y_0_*i*) Fixed 3.5	
	Log PFU/g	Y_end_	=average(Y_end_*i*) Fixed 3.2	
		LN(q)	=LN(1/(EXP(h_0_) − 1))	
Death rate	Log PFU/g/h	DR_mark_	=−0.0035 − 0.0001 × Temp_mark_	
Hepatitis A virus death	Log PFU/g	C1	=LogIC + 1/(1 − (10^-lY0-Yendl^/LN(10))) × (1 + EXP(-LN(q)) × DR_mark_ × Time_mark_	
TRANSPORTATION FROM MARKET TO HOME				
Transportation time	h	Time_veh_	=Uniform(0.325, 1.643)	[30]
Temperature during transportation	°C	Temp_veh_	=Pert(10.0, 18.0, 25.0)	[30]
Hepatitis A virus death				
		h_0_	=0.001	This research; [19]
	Log PFU/g	Y_0_	=average(Y_0_*i*) Fixed 3.5	
	Log PFU/g	Y_end_	=average(Y_end_*i*) Fixed 3.2	
		LN(q)	=LN(1/(EXP(h_0_) − 1))	
Death rate	Log PFU/g/h	DR_veh_	=−0.0035 − 0.0001 × Temp_veh_	
Hepatitis A virus death	Log PFU/g	C2	=C1 + 1/(1 − (10^-lY0-Yendl^/LN(10))) × (1 + EXP(-LN(q)) × DR_veh_ × Time_veh_	
HOME				
Storage time	h	Time_home_	=Uniform(0, 720)	[31]
Temperature during storage	°C	Temp_home_	=LogLogistic(−29.28, 33.22, 26.66, RiskTruncate(−5, 10))	[32]
Hepatitis A virus death				
		h_0_	=0.001	This research; [19]
	Log PFU/g	Y_0_	=average(Y_0_*i*) Fixed 3.5	
	Log PFU/g	Y_end_	=average(Y_end_*i*) Fixed 3.2	
		LN(q)	=LN(1/(EXP(h_0_)−1))	
Death rate	Log PFU/g/h	DR_home_	=−0.0035 − 0.0001 × Temp_home_	
Hepatitis A virus death	Log PFU/g	C3	=C2 + 1/(1 − (10^-lY0-Yendl^/LN(10))) × (1 + EXP(-LN(q)) × DR_home_ × Time_home_	
	PFU/g	C3_PFU/g_	=10^C3^	
CONSUMPTION				
Average consumption amount per day	g	Consump	=Expon (8.1689, RiskShift(0.69235), RiskTruncate(3, 36.3))	This research; [9]
Consumption ratio per day	%	ConRatio	=0.13	This research; [9]
		CR(0)	=1 − 0.13/100	[9]
		CR(1)	=0.13/100	[9]
		CR	=Discrete({0,1}, {CR(0), CR(1)})	[9]
	g	Amount	=IF(CR = 0, 0, Consump)	[9]
DOSE–RESPONSE				
Hepatitis A virus amount	PFU/g	D	=C3_PFU/g_ × Amount	
		*α*	=Fixed 0.373	[34]
		*β*	=Fixed 186.4	
RISK				
Probability of illness		P	=1 − (1 + D/*β*)^−*α*^	[34]

PR, prevalence; CL, contamination level; Temp, Temperature; DR, death rate; Consump, consumption; ConRatio, consumption ratio; D, dose.

## Data Availability

All related data and methods are presented in this paper. Additional inquiries should be addressed to the corresponding author.

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
