# Peer review of "Quantitative Risk Assessment of Hepatitis a Virus Infection Arising from the Consumption of Fermented Clams in South Korea"

_foods, 2023, doi:10.3390/foods12040796_

Round 1

Reviewer 1 Report

Please edit your manuscript in accordance with my suggestions and comments found in the revised communication file that is attached.

To accurately and fluently convey your findings, the entire manuscript will require strong English editing.

For better organisation, the abstract, methodology, introduction, results, and discussion sections should be improved.

Your ability to communicate your idea was hampered by the poor application of English, which led to confusion in many sentences.

You were not fortunate in some expressions, such as mentioning that you developed a predictive model, but the correct expression was that you applied your data to Baranyi predictive models to describe the kinetic behaviour of the HAV in fermented clams and were successfully fitted and the model predicted their fate.

Author Response

Response to Reviewer 1 Comments

Please edit your manuscript in accordance with my suggestions and comments found in the revised communication file that is attached.

To accurately and fluently convey your findings, the entire manuscript will require strong English editing.

For better organisation, the abstract, methodology, introduction, results, and discussion sections should be improved.

Your ability to communicate your idea was hampered by the poor application of English, which led to confusion in many sentences.

You were not fortunate in some expressions, such as mentioning that you developed a predictive model, but the correct expression was that you applied your data to Baranyi predictive models to describe the kinetic behaviour of the HAV in fermented clams and were successfully fitted and the model predicted their fate.

Point 1: Please clearly separate the aim, methodology, and findings sections within the abstract.

Response 1: The authors revised the abstract (Line 13-31).

Point 2: Line 18. Please make it clear that there are various stages, including transportation, etc.

Response 2: The authors described which data was used to apply the Baranyi predictive models (Line 18).

Point 3: Line 26. replace with direct simple expression, such as regular consumers of fermented clams.

Response 3: The authors changed the expression from ‘those who consumed’ to ‘regular consumers of’ (Line 26).

Point 4: Transfer it to Line 39 after "HAV is known as a causative agent of foodborne illness in several countries, mainly 38 due to contamination with clams such as oysters and mussels, and fruits and vegetables 39 such as leafy vegetables and berries [5]."

Response 4: The authors transferred the sentence to Line 50-52.

Point 5: Line 54-55. Indicate in the methodology of abstract.

Response 5: The authors indicated the methodology in Line 14-15.

Point 6: Line 81. Please keep artificial HAV inoculation steps and their counting separate from the application of observed counts to a predictive model.

Response 6: The authors separated the paragraph into ‘2.3. HAV inoculation and titer enumeration in the fermented clams’ and ‘2.4. Predictive model development’.

Point 7: Line 91-93. revise for correct structure and language

Response 7: The authors revised the sentence (Line 91-93).

Point 8: Line 122. You did not describe this parameter, fixed adjustment function.

Response 8: The authors described the meaning of the parameter (Line122).

Point 9: More explanation is needed to fully understand current applied models, so I proposed inserting the following paragraph.

Because HAV does not grow in food, the entire current experimental period that inoculated HAV undergoes in fermented clams is considered an inconstant lag phase, or adjustment process. Thus, the Combase model will simply estimate HAV survival quantity after exposure to various temperatures along the three different counting points, market, transportation, and home storage. As a result, the potential growth, p(t), variable cannot be used here; however, as previously stated by Barany, during the lag phase, the potential growth, p(t) variable is replaced by cells' actual physiological state, which is regarded as a transformation of the quantity q0.

Response 9: The authors appreciate your productive comment. In our study, the Baranyi model was used for the death curve rather than growth curve. Thus, your comment will be considered when we use the model for the bacterial growth.

Point 10: Line 100. You did not create the model, but rather applied your data to the Baranyi model provided by Combase. So, please correct expressions.

Response 10: The authors corrected the sentences (Line 125)

Point 11: Line 162-167. Use simple language and avoid repeating methodology.

Response 11: The authors revised the sentences (Line 189-190).

Point 12: To accurately and fluently convey your findings, the entire manuscript will require strong English editing.

Response 12: The whole manuscript was English-edited, and the authors attached the editing certificate.

Point 13: Table 2. Because your communication will be read by both regular readers and specialists, the table contains a number of abbreviations that must be written out in full for easy comparison and comprehension.

Response 13: The authors spelled out the abbreviations below the Table 2.

Reviewer 2 Report

The authors used a predictive microbiology model for QRA of Hepatitis A Virus Foodborne Illness by Intake of Fermented Clams in South Korea; in the title they should mention that they used a predictive model;

The authors should say what their goal is: to give directions to health authorities for risk management? or warn customers?

To better understand the study, the authors should provide a brief description of the food product (cultivated?, wild?, production technology, etc., pH,);

Line 41-43: In 2019, hepatitis A infection was prevalent in Korea, and the number of hepatitis A patients in 2019 was 17,598, which is a 7-fold increase compared to the previous year ; The authors should include more information;

Line 120-121: HAV dose-response model was investigated using published studies to analyze the response of  HAV foodborne illness to a human after consumption of fermented clams. The authors should show which published studies they used;

Line 165-166: As a result of developing primary predictive models for the number of HAV with the Baranyi model according to temperature and time change, the viral titers slightly decreased over time at 25°C. The authors should briefly discuss this.

Conclusions

The authors should better discuss the results for risk management purposes.

Author Response

Response to Reviewer 2 Comments

Point 1: The authors used a predictive microbiology model for QRA of Hepatitis A Virus Foodborne Illness by Intake of Fermented Clams in South Korea; in the title they should mention that they used a predictive model;

Response 1: Because predictive model was just part of this research (quantitative microbial risk assessment), adding quantitative microbial risk assessment might be appropriate rather than predictive model. Thus, the authors rewrote the title.

Point 2 : The authors should say what their goal is: to give directions to health authorities for risk management? or warn customers?

Response 2: The authors described the goal of this study (Line 68).

Point 3 : To better understand the study, the authors should provide a brief description of the food product (cultivated?, wild?, production technology, etc., pH,);

Response 3: The authors described the general information of the fermented clams (Line 60-62).

Point 4: In 2019, hepatitis A infection was prevalent in Korea, and the number of hepatitis A patients in 2019 was 17,598, which is a 7-fold increase compared to the previous year ; The authors should include more information;

Response 4: The authors described the global incidence of hepatitis A infection (Line 53-54).

Point 5: HAV dose-response model was investigated using published studies to analyze the response of HAV foodborne illness to a human after consumption of fermented clams. The authors should show which published studies they used;

Response 5: The authors described HAV dose-response model in the result section (Line 237-240).

Point 6 : As a result of developing primary predictive models for the number of HAV with the Baranyi model according to temperature and time change, the viral titers slightly decreased over time at 25°C. The authors should briefly discuss this.

Response 6: The authors discussed the reason why viral titers decreased at 25°C (Line 202-205).

Point 7 : Conclusions

The authors should better discuss the results for risk management purposes.

Response 7: The authors indicated the greatest correlated factor related to HAV foodborne illness and what consumers should be aware of (Line 267-269).

Round 2

Reviewer 1 Report

Please edit your manuscript in accordance with my suggestions and comments found in the revised communication file that is attached.

Dear authors, did you read my comment carefully? Your response "In our study, the Baranyi model was used for the death curve rather than the growth curve", suggests that you did not. You did not indicate in your methodology that the applied Baranyi model is for the death curve, so carefully read my suggestion and precisely describe the used model.

Furthermore, the term shoulder is always used in growth models to describe the rising curve, so I proposed more accurate terms such as survival curve.

Author Response

Response to Reviewer 1 Comments

Please edit your manuscript in accordance with my suggestions and comments found in the revised communication file that is attached.

Dear authors, did you read my comment carefully? Your response "In our study, the Baranyi model was used for the death curve rather than the growth curve", suggests that you did not. You did not indicate in your methodology that the applied Baranyi model is for the death curve, so carefully read my suggestion and precisely describe the used model.

Response : Thank you for your comments. The authors described the Baranyi model applied to the death curve model (Line 121-127).

Furthermore, the term shoulder is always used in growth models to describe the rising curve, so I proposed more accurate terms such as survival curve.

Point 1: These words are immediately followed by your description, but they are not mentioned in the middle of the sentence or followed by follow-stops. Please revise structure of the sentence to prevent confusion.

Response 1: The authors revised the sentence (Line 113-114).

Point 2: Insert ‘and stored at 15ºC’ .

Response 2: The authors inserted the phrase in line 132.

Point 3: Repeated, please indicate to figure. you can use;

The temperature and time-dependent changes in HAV contamination levels in fermented clams are depicted in Figure ? as described by the predictive models

Response 3: The authors presented primary predictive models to account for changes in HAV contamination levels as time and temperature changed (Figure 1).

Point 4: These words are immediately followed by your description, but they are not mentioned in the middle of the sentence or followed by follow-stops. Please revise structure of the sentence to prevent confusion.

Response 4: The authors deleted the phrase (Line 200).

Point 5: Confusing, replace with

The generated curves of primary predictive models for the number of HAV with the Baranyi model based on temperature and time changes revealed that viral titers at 25 °C decreased slightly over time.

Response 5: The authors changed the sentence as you guided (201-203).

Point 6: I think survival period would be more expressive.

Response 6: The authors cited the Baranyi model for the death curve with the reference of Baranyi and Roberts (1994) and a study by Baranyi et al. (1996) described the deaths curve with the Baranyi model. In this paper, the term ‘shoulder’ period was used for the death curve on page 1,030. In addition, as microbiological term, ‘shoulder’ was defined as the period time before the number of living microorganisms, begin to decrease by Perni et al. (2013). Thus, the authors used ‘shoulder period’ (Line 214).

Reference

Baranyi, J., & Roberts, T. A. (1994). A dynamic approach to predicting bacterial growth in food. International journal of food microbiology, 23(3-4), 277-294.

Baranyi, J., Jones, A., Walker, C., Kaloti, A., Robinson, T. P., & Mackey, B. M. (1996). A combined model for growth and subsequent thermal inactivation of Brochothrix thermosphacta. Applied and environmental microbiology, 62(3), 1029-1035.

Perni, S. (2013). Microbial control and safety in inhalation devices. In Inhaler devices (pp. 51-74). Woodhead Publishing.
